# An Evaluation of the In Vitro Roles and Mechanisms of Silibinin in Reducing Pyrazinamide- and Isoniazid-Induced Hepatocellular Damage

**DOI:** 10.3390/ijms21103714

**Published:** 2020-05-25

**Authors:** Zhang-He Goh, Jie Kai Tee, Han Kiat Ho

**Affiliations:** 1Department of Pharmacy, Faculty of Science, National University of Singapore, Singapore 117543, Singapore; goh.zhanghe@u.nus.edu (Z.-H.G.); a0072441@u.nus.edu (J.K.T.); 2NUS Graduate School for Integrative Sciences & Engineering, Centre for Life Sciences, National University of Singapore, Singapore 119077, Singapore

**Keywords:** drug-induced liver injury (DILI), silibinin, oxidative stress, tuberculosis, pyrazinamide, isoniazid

## Abstract

Tuberculosis remains a significant infectious lung disease that affects millions of patients worldwide. Despite numerous existing drug regimens for tuberculosis, drug-induced liver injury is a major challenge that limits the effectiveness of these therapeutics. Two drugs that form the backbone of the commonly administered quadruple antitubercular regimen, that is, pyrazinamide (PZA) and isoniazid (INH), are associated with such hepatotoxicity. Yet, we lack safe and effective alternatives to the antitubercular regimen. Consequently, current research largely focuses on exploiting the hepatoprotective effect of nutraceutical compounds as complementary therapy. Silibinin, a herbal product widely believed to protect against various liver diseases, potentially provides a useful solution given its hepatoprotective mechanisms. In our study, we identified silibinin’s role in mitigating PZA- and INH-induced hepatotoxicity and elucidated a deeper mechanistic understanding of silibinin’s hepatoprotective ability. Silibinin preserved the viability of human foetal hepatocyte line LO2 when co-administered with 80 mM INH and decreased apoptosis induced by a combination of 40 mM INH and 10 mM PZA by reducing oxidative damage to mitochondria, proteins, and lipids. Taken together, this proof-of-concept forms the rational basis for the further investigation of silibinin’s hepatoprotective effect in subsequent preclinical studies and clinical trials.

## 1. Introduction

Tuberculosis is an infectious lung disease caused by *Mycobacterium tuberculosis*, with one in four people affected globally [1]. While new and current drug regimens have been tailored to shorten treatment duration and increase therapeutic efficacy [2], drug-induced liver injury (DILI) caused by anti-tubercular therapy (ATT) still warrants the most concern. ATT has been reported to cause severe hepatotoxicity [3,4], which leads to discontinuation of therapy in 11% of patients [5]. Hence, the DILI arising from ATT is a pressing problem that needs to be addressed in tuberculosis.

The most prevalent ATT, known as the HRZE regimen, involves the use of four drugs in combination: pyrazinamide (PZA), isoniazid (INH), ethambutol (EMB), and rifampicin (RMP). Among them, PZA and INH are the most commonly implicated in DILI. PZA increases the risk of DILI by 3.5 times [6]. In HepG2, PZA-induced hepatotoxicity has been reported to invoke damage to cellular and mitochondrial membranes, leading to increased apoptotic activity in vitro [7]. Similarly, INH-induced hepatotoxicity occurs in approximately 25% of patients [8], and severe INH-induced hepatotoxicity occurs in 1 per 1000 patients [9]. At the same time, rat studies have also been conducted on PZA, INH, and RMP, as well as various combinations of these drugs. Administration of these combinations has led to increased membrane lipid peroxidation levels [10], increased serum levels of liver enzymes [11,12], and reduced antioxidant protein levels [12,13]. Notably, the search for strategies to reduce PZA- and INH-induced hepatotoxicity is further underscored by the first-line status of the HRZE regimen in ATT with few safer and equally efficacious alternatives [14,15].

The two main mechanisms through which PZA and INH injure hepatocytes are both associated with oxidative stress [16]. First, PZA and INH can be converted to reactive metabolites by drug metabolizing enzymes. PZA is oxidized to 5-hydroxypyrazinamide via xanthine oxidase; and both PZA and 5-hydroxypyrazinamide are further bioactivated by xanthine oxidase to the toxic metabolites, pyrazinoic acid and 5-hydroxypyrazinoic acid [17,18]. Between pyrazinoic acid and 5-hydroxypyrazinoic acid, 5-hydroxypyrazinoic acid may be the more toxic metabolite [18]. In contrast, INH can be activated to toxic metabolites via N-acetyltransferase [8,19,20], amidases [8,19,20], and CYP2E1 [8,19]. Notably, INH also induces CYP2E1 [21,22], thus increasing the rate at which INH is metabolised. These metabolites of INH increase the levels of intracellular reactive oxygen species (ROS) [8,23,24], which consequently damage vital cellular targets. The subcellular consequences include the following: deoxyribonucleic acid (DNA) fragmentation [24], lipid peroxidation [24,25,26], and protein carbonylation [20,26]. Therefore, given that the multifactorial nature of DILI arising from ATT includes interindividual variability in metabolic processes, patients may exhibit features of idiosyncratic toxicity.

Second, PZA and INH suppress the nuclear factor (erythroid-derived 2)-like 2 (Nrf2)-antioxidant response element (ARE) pathway that protects cells from oxidative damage [27,28]. Consequently, both INH [27] and PZA [28] increase cellular susceptibility to oxidative stress by decreasing the expression of downstream antioxidant proteins; the antioxidant proteins affected include glutamate-cysteine ligase catalytic subunit (Gclc), NAD(P)H quinone dehydrogenase 1 (NQO1), heme oxygenase-1 (HO-1), and sulfiredoxin 1 (Srxn1). Indeed, RMP, INH, and PZA have been shown to potentiate the hepatotoxic effects of one another in in vitro assays involving HepG2 [29], though the underlying mechanisms of hepatotoxicity caused by these antitubercular drugs remain poorly understood to date. Seen in totality, the broad strokes illustrated by these studies denote the need for hepatoprotective strategies to counter the increase in oxidative stress induced by these drugs.

The most well-researched strategy to reduce HRZE-induced hepatotoxicity involves the use of antioxidant nutraceuticals to reduce oxidative stress and liver inflammation [30]. Among the nutraceuticals that have been explored for their hepatoprotective potential, silibinin is the gold standard [31]. Silibinin is a herbal product derived from milk thistle that has been postulated to protect against liver injury caused by various chemotherapeutic [32] and toxic agents [14,33,34,35]. Two factors contribute to silibinin’s popularity over other chemical drugs in liver disease: it has low toxicity [36] and exhibits a broad spectrum of hepatoprotective mechanisms [37,38,39,40,41]. Many of silibinin’s mechanisms of action can be attributed to its antioxidant, anti-inflammatory, immunomodulatory, and antifibrotic actions [42]. Silibinin’s anti-inflammatory and immunomodulatory effects are manifested through silibinin’s actions on pathways involving tumor necrosis factor-alpha (TNF-α) [43] and NF-κB [44], as well as its modulation of lipopolysaccharide-induced NO production [44] and NOD-like receptor pyrin domain-containing-3 (NLRP3) inflammasome activation [45]. At the same time, silibinin’s antioxidant effect has been attributed to its ability to inhibit ROS-producing enzymes, directly scavenge free radicals, prevent the absorption of ions by the intestine through chelation, and promote the expression of protective molecules and enzymes that mitigate oxidative stress [31,42]. Therefore, silibinin may be uniquely placed to mitigate the principal mechanisms implicated in oxidative stress responsible for HRZE-induced hepatotoxicity.

Unfortunately, despite the extensive and rigorous research on the basis for silibinin’s hepatoprotective effect in recent years [5,31,46], the in vivo and in vitro biological markers that silibinin modulates have not been conclusively linked to the reduction of DILI [47]. Furthermore, the exact biochemical mediators behind silibinin’s hepatoprotective effect have also not been identified [48]. Silibinin’s hepatoprotective nature remains nebulous to date, making it especially challenging to clarify and optimise silibinin’s role in mitigating HRZE-induced hepatotoxicity. Consequently, silibinin’s reduction of HRZE-induced hepatotoxicity has neither been definitively proven nor characterised [49,50,51]. Therefore, a deeper mechanistic understanding of silibinin’s hepatoprotective ability must be elucidated before silibinin can be widely used as an adjuvant to ameliorate HRZE-induced hepatotoxicity.

In this study, we sought to investigate the role of silibinin in mitigating PZA- and INH-induced hepatotoxicity. We hypothesised that silibinin reduces PZA- and INH-induced hepatotoxicity through its anti-oxidative mechanisms. Indeed, our results showed that silibinin preserved cell viability when co-administered with INH. We also determined that the co-administration of silibinin with a combination of INH and PZA (I/P) led to a reduction in oxidative damage to intracellular targets and apoptotic activity. Together, these findings supported our hypothesis that silibinin reduces PZA- and INH-induced hepatotoxicity through its modulation of oxidative stress.

## 2. Results

### 2.1. Silibinin Mitigated Hepatotoxicity Induced by INH When Administered as a Rescue Adjuvant

As silibinin’s hepatoprotective role is often discussed in conjunction with its anticancer and antiproliferative properties [52,53,54], we first optimised silibinin’s treatment duration and established a suitable range of concentrations of silibinin that could be used safely without precipitating adverse effects. By testing the effects of various concentrations of silibinin on cell viability over 72 h, we determined silibinin’s maximum non-toxic concentration to be 50 μM (Appendix A). Consequently, subsequent experiments focused on testing silibinin’s hepatoprotective effect at the concentrations of 25 μM and 50 μM. Similarly, to optimise the concentration windows of INH and PZA, we determined their half maximal inhibitory concentration (IC50) to be 73 and 60 mM, respectively (Appendix A), observations consistent with reports in those made in HepaRG [47] and HepG2 [29]. We define these observations as the “Goldilocks zone” (i.e., synonymous to a zone that is neither too high nor too low)—in which we expected to observe silibinin’s hepatoprotective effect.

Having established the optimal concentrations of silibinin, PZA, and INH to be used in our experiments, we then profiled silibinin’s orthogonal roles as a preventive, rescue, or recovery adjuvant in reducing PZA- and INH-associated hepatotoxicity. This approach was based on differential sequencing of the toxicant and silibinin exposure (Figure 1). In our exploration of silibinin as a recovery adjuvant, we investigated silibinin’s ability to mitigate hepatocyte toxicity in vitro after hepatotoxic induction. We set up a pair of experiments to demonstrate this in vitro. In the first experiment, the hepatocytes underwent a washout procedure, where we replaced the toxicant media with fresh media after 24 h. In the second experiment, there was no washout and the toxicant remained in the culture medium. By comparing silibinin’s in vitro hepatoprotective ability between this pair of experiments, we simulated silibinin’s potential ability to mitigate further liver injury in patients who either discontinue or stay on the hepatotoxic regimen.

Three major observations can be made about our experiments that serve to identify silibinin’s role in protecting against DILI. First, silibinin was effective in rescue (Figure 2A), but not in prevention and recovery (Figure 2B,C). The co-administration of 25 μM silibinin with 80 mM INH moderately protected against INH-induced hepatotoxicity, raising the mean hepatocyte viability from 53% to 63% (Figure 2A). Second, when hepatotoxicity was induced by a higher concentration of INH at 100 mM, the magnitude of silibinin’s hepatoprotective effect decreased slightly and silibinin’s optimal hepatoprotective concentration rose to 50 μM (Figure 2A). Inducing hepatotoxicity using a lower concentration of INH at 50 mM also appeared to negate silibinin’s hepatoprotective effect (Figure 2A). Third, silibinin’s hepatoprotective effect was independent of PZA-induced hepatotoxicity in vitro (Figure 2A–C). Interestingly, silibinin’s protection against INH-induced hepatotoxicity also seemed to decrease when silibinin was administered together with a combination of 50 mM INH and 50 mM PZA (Figure 2A). This observed combinatorial toxic effect between INH and PZA is reminiscent of the synergistic toxicity between the four drugs found in the HRZE regimen, which has been reported in vitro [29,55,56], in vivo [57,58], and in humans [59,60]. Everything considered, these results suggest that most of silibinin’s hepatoprotective effect may be the most apparent at moderate levels of INH-mediated toxicity.

### 2.2. Silibinin Reduced Oxidative Damage of INH and PZA on Classical Intracellular Targets

After establishing silibinin’s role as a rescue adjuvant in INH-induced hepatotoxicity, we characterised silibinin’s ability to reduce intracellular ROS levels and oxidative damage to proteins, lipids, and DNA. We assessed these intracellular indicators of oxidative stress for two reasons: they play critical roles in cellular function and survival, and their measurements have been widely studied and are well-established [61]. These experiments showed that 50 μM silibinin mitigated the increase in intracellular ROS levels when co-administered with I/P 40/10 over 24 h (Figure 3A).

To assess whether the attenuation of intracellular ROS production translated into a reduction in damage to important biomolecules, we then quantified the corresponding oxidative damage incurred on proteins, lipids, and DNA. Specifically, we quantified the oxidative damage through protein carbonylation levels, lipid peroxidation levels, and DNA fragmentation. These experiments revealed that 25 and 50 μM silibinin significantly reduced protein carbonylation and lipid peroxidation levels (Figure 3B,C). Importantly, silibinin’s reduction of oxidative stress was independent of DNA oxidative damage induced by I/P 40/10 (Figure 3D), as measured using the Comet assay, which is especially useful for detecting genotoxicity because it paints a holistic picture of overall DNA damage by accounting for multiple genotoxic mechanisms [62]. Overall, silibinin’s reduction of ROS levels led to a reduction in protein carbonylation and lipid peroxidation, but not in DNA fragmentation.

### 2.3. Silibinin Protected against Apoptosis by Maintaining Mitochondrial Membrane Potential

As oxidative stress has been reported to trigger apoptosis via caspase-9 and, subsequently, caspase-3 activation in the intrinsic pathway [63], the drug-induced ROS levels would likely result in an increase in cell death as well. Indeed, silibinin has been reported to reduce caspase-3 activation [40]. Therefore, we measured silibinin’s ability to reduce apoptotic activity in LO2 to further reinforce the association between the observed decrease in cell viability and the increase in ROS levels. The use of caspase-3 activity to gauge apoptosis yielded two key observations (Figure 4A). First, the co-administration of silibinin significantly mitigated the induction of caspase-3 activity by I/P 40/10. As caspase-3 is the final mediator of both the intrinsic and extrinsic apoptotic pathways, silibinin’s mitigation of caspase-3 induction suggests that it reduces the overall apoptotic activity induced by I/P 40/10 administration. Second, the administration of silibinin alone did not affect caspase-3 activity. This suggests that the observed decrease in cell viability is solely attributed to the exposure of LO2 to I/P 40/10.

Oxidative mitochondrial stress has been identified as the key driver of INH-induced apoptotic activity; the increase in ROS levels promotes megamitochondria formation, consequently triggering cytochrome c release and upregulating apoptotic signalling [23,64]. Thus, having observed that silibinin attenuated apoptotic activity, we then interrogated silibinin’s effect on preserving mitochondrial function. We observed that silibinin indeed slightly attenuated the proportion of cells with mitochondrial membrane potential transition induced by I/P 40/10 (Figure 4B). This may suggest that silibinin’s reduction of oxidative stress may ameliorate mitochondrial dysfunction and, in turn, apoptotic activity.

### 2.4. Silibinin Restored HO-1 Expression and Induced Srxn1 Expression in Transforming Growth Factor-α Transgenic Mouse Hepatocytes (TAMH)

Another aspect of INH’s hepatotoxic effect arises from its suppression of proteins expressed in the Nrf2–ARE pathway [27,28]. However, this effect has not been reported for PZA. To evaluate whether silibinin’s hepatoprotective effect entails the induction of Nrf2–ARE-related protein expression, we first profiled the individual effects of INH and PZA in LO2. In these preliminary tests, INH, but not PZA, suppressed the expression of HO-1. However, other ARE responsive genes, such as Gclc, NQO1, and Srxn1, were not suppressed by INH (Appendix A). Silibinin was then co-administered with I/P 40/10 to test our hypothesis that silibinin’s utility in the HRZE regimen arises from its induction of these antioxidant enzymes. Because the expressions of these four antioxidant proteins may differ across cell lines, we tested our hypothesis in both LO2 and TAMH.

While silibinin has been reported to exhibit an indirect antioxidative effect by upregulating the Nrf2–ARE pathway [65], we found that silibinin’s hepatoprotection in LO2 was independent of Gclc, HO-1, NQO1, and Srxn1 induction (Figure 5A). Silibinin alone did not induce the expression of Gclc, HO-1, NQO1, and Srxn1, suggesting that it does not perturb endogenous oxidative stress response, a finding indicative of silibinin’s safety. As the Nrf2–ARE pathway in LO2 may not be sensitive to suppression by I/P compared with other cell lines, we further verified our observation in TAMH, in which the administration of silibinin alone induced Srxn1 expression. Moreover, when co-administered with 40 mM INH, silibinin restored HO-1 expression to normal levels, though these effects were independent of Gclc and NQO1 induction (Figure 5B).

## 3. Discussion

DILI is the most frequently cited reason for the withdrawal of drugs, especially when the manifestations of the hepatotoxicity are complex and require a better understanding of the underlying mechanisms [66]. Therefore, we have simulated silibinin’s clinical roles in prophylaxis, rescue, and recovery of HRZE-induced hepatotoxicity using an in vitro model with the respective pre-, co-, and post-administration of silibinin with the hepatotoxic regimens. As a prophylactic agent, silibinin would be taken before starting the HRZE regimen to protect patients from future hepatotoxicity; as a rescue agent, silibinin would be co-administered with the HRZE regimen to mitigate hepatotoxicity; and as a recovery agent, silibinin would be prescribed after the onset of HRZE-induced hepatotoxicity to aid in the healing process. We found that silibinin was mainly useful as a rescue adjuvant (Figure 2A) and ascertained that silibinin’s hepatoprotective effect arises from two aspects. First, silibinin reduces intracellular levels of oxidative stress and oxidative damage to intracellular targets (Figure 3A–D) and mitochondria (Figure 4B), leading to decreased apoptotic activity (Figure 4A). This observation is consistent with silibinin’s ability to reduce in vivo markers of direct oxidative damage in human hepatocytes, such as DNA fragmentation levels, lipid peroxidation, and mitochondrial dysfunction, also reported by other authors [31,32,39,67]. Second, silibinin induces Nrf2–ARE-related protein expression (Figure 5B). This also coincides with silibinin’s ability to increase levels of endogenous proteins that protect cells from oxidative damage [68], including various mediators along the mitogen-activated protein kinase (MAPK) pathway [69], thioredoxin [38], and superoxide dismutase (SOD) [70].

When used as a rescue adjuvant, silibinin was the most significantly hepatoprotective within INH’s “Goldilocks zone” (Figure 2A). Specifically, the “Goldilocks zone” is a range of toxicant concentrations around its IC50, the toxicant concentrations that trigger DILI to approximately 50% cell viability: the IC50 of PZA and INH is 60 and 73 mM respectively (Appendix A). This implies that silibinin may be the most efficacious within a specific window of INH concentrations that are neither too severe nor too mild. This observation is consistent with other in vitro findings, which involve the characterisation of silibinin’s hepatoprotective effects at toxicant concentrations around the IC50 values of their respective assays [29,47]. The existence of INH’s “Goldilocks zone” has two clinical implications in discussing silibinin’s role as a rescue adjuvant. First, it suggests that silibinin may be the most efficacious at moderate levels of DILI, and correspondingly less hepatoprotective in very early or late stages of DILI. Thus, depending on a patient’s liver function, silibinin’s dose can be carefully titrated to optimise the magnitude of hepatoprotection, while reducing silibinin’s potential side effects. The optimal silibinin concentration from our viability experiments was determined to be 25 μM when DILI was induced with 80 mM INH (Figure 2A). This may be explained by silibinin’s pro-oxidative and pro-apoptotic effects at higher concentrations (i.e., beyond 100 µM) (Appendix A), an observation that corroborates experiments conducted by other research groups in rats [71] and other in vitro cell lines [31,47,72]. Our results and literature reports [31,47,71,72] thus suggest that increasing silibinin’s concentration may not always lead to increased hepatoprotection.

At the same time, silibinin was not useful as an adjuvant in prophylaxis and recovery (Figure 2B,C). The lack of prophylactic effect may suggest that silibinin may not prevent HRZE-induced hepatotoxicity (Figure 2B). We also observed that silibinin did not promote the recovery process (Figure 2C), suggesting that silibinin may also not help patients recover from HRZE-induced hepatotoxicity. While these two observations suggest that silibinin may not be involved in the protective or regenerative mechanisms that restore normal hepatocyte function after DILI has occurred [73], silibinin has been shown to reduce stellate cell migration [57,65,74], which plays an important role in mediating liver diseases involving fibrotic activity, liver injury, and liver regeneration [75].

Therefore, future directions to better characterise silibinin’s role in recovery may centre around the use of co-cultures, which can mimic paracrine responses. These studies will help clinicians personalize regimens to patients’ conditions rapidly and accurately; instead of using a one-size-fit-all approach [76], regimens may be tailored to patients depending on their risk of DILI from genetic polymorphisms [68]. Ethical considerations in clinical practice often require investigators to exclude moderate-to-high-risk patients in their study [51], and consequently, investigators may not recruit enough patients to power their study more effectively [49]. Our finding that silibinin was most effective in moderate-to-high DILI (Figure 2A) may establish the moral basis for further clinical trials investigating silibinin’s hepatoprotective effect.

Silibinin also protected against apoptosis induced by I/P (Figure 4A) independently of viability restoration (Figure 2A) by reducing intracellular oxidative stress. This reduction in I/P-induced oxidative stress manifested in two ways: decreased oxidative damage to classical intracellular targets, as measured by protein carbonylation (Figure 3B) and lipid peroxidation (Figure 3C), and in the restoration of mitochondrial membrane potential (Figure 4B). Interestingly, though silibinin has been reported to protect against doxorubicin-induced DNA oxidative fragmentation in mice [32], silibinin did not appear to protect against oxidative DNA damage induced by I/P 40/10 in our study (Figure 3D). Our observations on these four hallmarks of oxidative damage corroborate existing in vitro and in vivo studies on silibinin’s antioxidant effect in showing that silibinin mitigates the elevated lipid peroxidation and protein carbonylation levels in DILI [77], and further imply that silibinin may not protect against all forms of INH- and PZA-induced oxidative damage. At the same time, our Comet assay results lend credence to silibinin’s safety at the concentrations used in our study. Our observation that there was no difference in the magnitudes of the tail moment and olive moment between the control and silibinin-treated samples (Figure 3D) suggests that silibinin did not induce DNA damage in our study. Coupled with the observation that silibinin did not induce caspase-3 activity (Figure 4A), this reinforces silibinin’s clinical safety profile [51,78] and supports silibinin’s development in further studies.

Other than functioning as a direct antioxidant, silibinin and its analogues have also been reported to induce the levels and activities of various endogenous antioxidants in hepatocytes [46,65]. Therefore, we chose to investigate Nrf2–ARE, a major antioxidant pathway. Silibinin’s hepatoprotective effect in LO2 was independent of Nrf2–ARE pathway activation or restoration after suppression by INH (Figure 5A), agreeing with our earlier observation that silibinin did not function as a preventive agent in vitro (Figure 2B); if silibinin had adequately induced the protective Nrf2–ARE pathway, the upregulation of antioxidant enzymatic systems would serve to protect against INH- or PZA-induced hepatotoxicity. The observation that INH suppressed HO-1 expression in LO2 when used in combination with PZA (Figure 5A) is consistent with the current paradigm in which INH reduces both the mRNA transcription levels and the activity of HO-1 [27,28]. By suppressing HO-1 expression, INH increases LO2 cells’ susceptibility to oxidative stress mediated by the increase in intracellular ROS levels (Figure 3A).

In contrast, silibinin induced Srxn-1 expression and restored HO-1 expression in TAMH (Figure 5B). Interestingly, though we found that this was independent of silibinin’s hepatoprotection in TAMH (Appendix A), our findings corroborate evidence in rats [65] and mice [79] that silibinin’s induction of the Nrf2–ARE pathway may contribute towards its hepatoprotective effect in rodents. Moreover, because INH suppressed Gclc, NQO1, and Srxn1 expression in TAMH, but not in LO2, the observed decrease in viability in LO2 (Figure 2A) is likely independent of the expression levels of these three proteins. This also suggests that Gclc, NQO1, and Srxn1 may play a smaller role in mitigating oxidative stress induced in LO2 by I/P.

The differences in observations made in LO2 and TAMH can be ascribed to possible Nrf2-independent mechanisms of hepatoprotection. Indeed, apart from exhibiting a direct hepatoprotective effect, silibinin may modulate pathways other than Nrf2-related upregulation of antioxidant enzymes. In fact, the hepatoprotective effect of tert-butylhydroquinone has been ascribed to its effects on autophagy [80], and a similar effect may exist in LO2. In contrast, the Nrf2-independent modulation of HO-1 expression has also been reported in cases of muscular atrophy [81]. Taken together, these observations suggest that ARE expression may also be controlled by less understood constitutive pathways besides the well-established induction by Nrf2 activation [82].

An alternative explanation for this interesting phenomenon is that the Nrf2–ARE pathway is activated by silibinin’s metabolites, rather than silibinin itself. As TAMH is metabolically active [83], it may convert silibinin to metabolites that structurally resemble the analogue 2,3-dehydrosilydianin, which has been reported to upregulate NQO1 activity [46]. The differences between our results and previous findings [46,65] may thus be attributed to innate metabolic, transporter-related, and physiological differences between various cell lines. Our choice of LO2 has its distinct advantages; not only is LO2 more representative of human liver physiology than HepG2 [84], but LO2 also expresses higher levels of CYP2E1 than HepaRG that enable LO2 to convert INH to its toxic metabolite hydrazine [85]. Notably, HepaRG’s poor expression of CYP2E1 has cast doubt on its relevance in INH-induced hepatotoxicity models [86]. Concurrently, our in vitro experiments using human cell lines serve as useful cross-references for other in vitro [7,47], animal in vivo [11], and human [49,50,51] studies on silibinin’s protection against HRZE-induced hepatotoxicity.

At the same time, we observed that silibinin was more hepatoprotective against INH than PZA or I/P. This observation appears to suggest that silibinin may not mitigate mechanisms involved in PZA-induced hepatotoxicity in vitro. Therefore, silibinin may need to be used carefully as a rescue adjuvant in the overall HRZE regimen, which combines the use of INH and PZA. In fact, silibinin may be more useful in triple ATT regimens that exclude the use of PZA, which are often used in patients who suffer from hepatotoxicity [87]. Indeed, PZA-induced hepatotoxicity is complex and poorly understood. Despite PZA’s greater association with hepatotoxicity than INH, research on hepatotoxicity has mostly centred on the latter, and the mechanisms responsible for INH-induced hepatotoxicity are becoming more well understood in recent years [88]. Specifically, oxidative stress arising from the toxic INH metabolite hydrazine has been validated as a major mechanism in INH-induced hepatotoxicity using pharmacodynamic and pharmacokinetic evidence [64,88,89]. In contrast, while several of PZA’s metabolites have been identified [18], research characterising their toxicities has only just started emerging [25]. Recently, 5-hydroxypyrazinoic acid, a metabolite of PZA, was proposed to be primarily responsible for PZA’s toxicity [18,25]. However, while the conversion of PZA to 5-hydroxypyrazinoic acid is mediated by xanthine oxidase, silibinin did not protect against PZA-induced hepatotoxicity in our study (Figure 2A–C), despite being a xanthine oxidase inhibitor [31,90]. One potential limitation of our study may lie with our finding that silibinin did not protect against PZA-induced hepatotoxicity, which may be attributed to other injury pathways [50]. Because these alternative pathways may only become apparent in vivo, they lie beyond the scope of our study.

## 4. Materials and Methods

### 4.1. Cell Culture and Reagents

LO2 is a human foetal hepatocyte cell line that has been previously characterised [91]. TAMH was a kind gift from the late Prof. Nelson Fausto (University of Washington, Seattle, WA, USA); the isolation of TAMH was previously described [92]. LO2 was cultured in Dulbecco’s minimum essential medium (DMEM) (Sigma Aldrich, St. Louis, MO, USA) containing 10% *v*/*v* foetal bovine serum (FBS). TAMH was cultured in DMEM-F12 (Sigma Aldrich, St. Louis, MO, USA). Cells were incubated at 37 °C in a humidified incubator with 5% CO_2_. Stocks of 100 mM silibinin (Sigma Aldrich, St. Louis, MO, USA), 10 mM sulphoraphane (SU) (Sigma Aldrich, St. Louis, MO, USA), 50 mM trans-cinnamaldehyde (CA) (Sigma Aldrich, St. Louis, MO, USA), and 5 M tert-butyl hydroperoxide (TBHP) (Sigma Aldrich, St. Louis, MO, USA) were prepared in dimethyl sulfoxide (DMSO) (Sigma Aldrich, St. Louis, MO, USA). Stocks were diluted with culture medium into different concentrations, ensuring that the final concentration of DMSO never exceeded 0.1% *v*/*v*.

### 4.2. Cell Viability Assay

First, 10,000 cells per well were seeded in a clear 96-well plate overnight and treated accordingly. At each timepoint, 0.5 mg/mL 3-(4,5-dimethylthiazol-2-yl)-2,5-diphenyltetrazolium bromide (MTT) solution in fresh media was added to the cells and incubated for 3 h at 37 °C. Thereafter, the MTT solution was removed, the formazan crystals formed dissolved in 200 μL DMSO, and the absorbance was measured at 570 nm using Hidex sense microplate reader (Hidex, Turku, Finland). Stocks of 100 mM silibinin (Sigma Aldrich, St. Louis, MO, USA) were prepared in dimethyl sulfoxide (DMSO) (Sigma Aldrich, St. Louis, MO, USA).

In measuring the toxicity of silibinin, INH, and PZA on cells, a series of concentrations was prepared in media and administered to silibinin for 24 h, after which the cell viability was measured. To simulate silibinin’s role as a preventive agent, the cells were treated with silibinin for 24 h, then with both silibinin and the toxicants for a further 24 h, before cell viability was assessed. To simulate silibinin’s role as a rescue adjuvant, the cells were treated with both silibinin and the toxicant for 24 h before cell viability was assessed. To simulate silibinin’s role as a recovery adjuvant without washout, the cells were treated with the toxicant for 24 h, then with silibinin and the toxicants for a further 24 h, before cell viability was assessed. To simulate silibinin’s role as a recovery adjuvant with washout, the cells were treated with the toxicant for 24 h, then with only silibinin for a further 24 h, before the cell viability was assessed.

### 4.3. Direct ROS Quantitation

First, 10,000 cells per well were seeded in a black 96-well plate overnight and treated accordingly. At each timepoint, cells were washed with phosphate-buffered saline (PBS) and incubated for 30 min with 10 µM 6-chloromethyl-2′,7′-dichlorodihydrofluorescein diacetate (DCFDA) and 1 mg/L Hoechst 33342 dye (Sigma Aldrich, St. Louis, MO, USA) diluted in media. Then, 100 µL of PBS was added and fluorescence was measured at λex/λem = 350/461 nm (Hoescht) and 485/535 nm (DCFDA), respectively, using a Hidex sense microplate reader (Hidex, Turku, Finland).

### 4.4. Lipid Peroxidation Quantitation

First, 20,000 cells/cm^2^ were seeded onto a 100 mm dish overnight and the respective treatment media was added. Thiobarbituric acid reactive substances (TBARS) were quantified using the TBARS assay kit (Cayman Chemical, Ann Arbor, MI, USA) following the manufacturer’s instructions. Briefly, both the adhered live cells and floating dead cells were harvested, resuspended in 150 µL PBS, and sonicated for 5 min. Then, 100 µL of resuspended pellet and 100 µL sodium dodecyl sulfate (SDS) solution were added to test tubes and mixed with 4 mL of colour reagent solution, which constituted 530 mg of 2-thiobarbituric acid dissolved in 50 mL of a 1:1 mixture of 20% *v*/*v* acetic acid and 10% *w*/*v* sodium hydroxide. The tubes were boiled for 1 h, immersed in ice for 10 min to quench further reaction, centrifuged at 2000× *g* for 10 min at 4 °C, warmed to room temperature, and added to a 96-well black plate. Fluorescence intensity was measured at λex/λem = 520/560 nm using a Hidex sense microplate reader (Hidex, Turku, Finland). Lipid peroxidation levels were normalised against total protein quantitated using the Pierce bicinchoninic acid (BCA) Protein Assay Kit (Thermo Fisher Scientific, Waltham, MA, USA).

### 4.5. Protein Carbonylation Quantitation

First, 20,000 cells/cm^2^ were seeded onto a 100 mm dish overnight and treated accordingly. Both live and dead cells were harvested and resuspended in 150 µL MilliQ Grade I water. Then, 100 µL of this suspension was added to 500 µL 10% *v*/*v* trichloroacetic acid (TCA) solution and centrifuged at 13,000 rpm for 2 min. The cell pellet was collected and incubated with 100 µL 0.02% *w*/*v* 2,4-dinitrophenylhydrazine (2,4-DNPH) hydrochloride (Tokyo Chemical Industries, Tokyo, Japan) for 1 h with constant vortexing. Then, 50 µL 100% *v*/*v* TCA was added and the suspension was centrifuged at 13,000 rpm for 5 min. The cell pellet was washed with cold acetone and dissolved in 200 µL 6M guanidine HCl (Sigma Aldrich, St. Louis, MO, USA). The absorbance of the solution was measured at 375 nm in a Hidex sense microplate reader (Hidex, Turku, Finland) to determine carbonyl levels, which were normalised against total protein quantitated using the Pierce BCA Protein Assay Kit (Thermo Fisher Scientific, Waltham, MA, USA).

### 4.6. Comet Assay

First, 400,000 cells per well were seeded on a 12-well plate overnight, treated accordingly, harvested, and resuspended in 100 µL PBS. Then, 20 µL of this suspension was mixed with low-melting agarose (Trevigen, Gaithersburg, MD, USA), spread evenly over CometSlides (Trevigen, Gaithersburg, MD, USA), left to congeal, and then kept in lysis solution (Trevigen) at 4 °C overnight. Thereafter, the slides were immersed in unwinding solution for 30 min at room temperature before gel electrophoresis was run for 25 min. The slides were washed, dried at 37 °C overnight, and stained with SYBRGold (Qiagen, Hilden, Germany). Fluorescence images were taken with Olympus Fluoview FV1000 confocal microscope (Olympus, Tokyo, Japan) and analysed using OpenComet [93]. The tail moment and the olive moment were calculated as follows:Tail Moment = Tail Length × Tail DNA%(1)
Olive Moment = Tail DNA% × distance between head and tail means(2)

### 4.7. Mitochondrial Membrane Potential Measurement

First, 20,000 cells/cm^2^ were seeded onto a 60 mm dish overnight and treated accordingly. They were then incubated with tetramethylrhodamine-methyl-ester (TMRM) (Thermo Fisher Scientific, Waltham, MA, USA) for 30 min, harvested, centrifuged, and reconstituted in 400 µL PBS. The percentage of cells within a defined range of fluorescence intensity was determined with Beckman Coulter CyAn Advanced Digital Processing (ADP) flow cytometer (Beckman Coulter, Jersey, NJ, USA).

### 4.8. Apoptosis Detection

First, 20,000 cells/cm^2^ were seeded onto a 60 mm dish overnight and treated accordingly. Cell lysates were extracted with radioimmunoprecipitation assay (RIPA) buffer containing 0.1% *w*/*v* SDS, 1% *w*/*v* NP-40, and 0.5% *w*/*v* sodium deoxycholate in PBS. Then, 300 µL of protease assay buffer (2 mM dithiothreitol (DTT), 10% *v*/*v* glycerol, 20 mM 4-(2-hydroxyethyl)-1-piperazineethanesulfonic acid (HEPES), and 20 mM Ac-DEVD-AMC caspase-3 fluorogenic substrate (BD Pharmingen, Franklin Lakes, NJ, USA) was added and the samples were incubated at 37 °C in the dark for 1 h. Then, 100 µL of each sample was added to a 96-well black plate and the fluorescence intensity was measured at λex/λem = 390/444 nm using a Hidex sense microplate reader (Hidex, Turku, Finland).

### 4.9. Western Blot

Cell lysates were extracted using RIPA buffer and protein concentrations were normalised using the Pierce BCA Protein Assay Kit (Thermo Fisher Scientific, Waltham, MA, USA). Proteins were mixed with loading dye, boiled at 100 °C for 5 min, separated on SDS-polyacrylamide gel (PAGE) using 12% *v*/*v* polyacrylamide gels (Bio-Rad Laboratories, Hercules, CA, USA), and then transferred onto polyvinylidene difluoride (PVDF) membrane (Thermo Fisher Scientific, Waltham, MA, USA) at 4 °C with 100 V for 2 h. Membranes were washed with tris-buffered saline (1st Base, Singapore) containing 0.1% *v*/*v* Tween, blocked with 5% *w*/*v* bovine serum albumin (BSA), and then incubated overnight at 4 °C with the following primary antibodies in 2% *w*/*v* BSA: rabbit anti-Gclc antibody (Abcam, Cambridge, UK; 1:1000); rabbit anti-NQO1 antibody (Cell Signalling, Danvers, MA, USA; 1:1000); rabbit anti-HO-1 antibody (Cell Signalling; Danvers, MA, USA; 1:1000); mouse anti-Srxn1 antibody (Santa Cruz, Dallas, TX, USA; 1:500); and mouse anti-β-actin antibody (Cell Signalling, Danvers, MA, USA; 1:10,000). Bound antibodies were detected using horseradish peroxidase-conjugated secondary antibodies and visualized by chemiluminescence using Western Lightning Plus-ECL reagent (Perkin Elmer, Waltham, MA, USA). Band intensities were analysed using ImageJ (National Institutes of Health, Bethesda, MD, USA) and normalised using β-actin.

### 4.10. Statistical Analysis

Statistical analysis was conducted using GraphPad Prism. The results were expressed as means ± S.E.M. Differences in mean values were analysed by t-tests or one-way analysis of variance (ANOVA) with Dunnett’s correction. A *p*-value < 0.05 was considered statistically significant.

## 5. Conclusions

In summary, we have assessed and characterised silibinin’s various roles as an adjuvant in protecting against PZA- and INH-induced hepatotoxicity. Our in vitro experiments suggest that silibinin may be safe and efficacious as a rescue adjuvant, both fundamental considerations in the use of any drug. Further optimisation of our in vitro model may also enhance silibinin’s hepatoprotective effect in rescue, prophylaxis, and recovery. Using this model, we have gleaned important mechanistic insights into its hepatoprotective effect and identified novel antioxidant targets in ameliorating HRZE-induced hepatotoxicity. Future directions will involve exploring the two main mechanisms by which silibinin may ameliorate hepatotoxicity; the proof-of-concept demonstrated in this project will inform subsequent in vitro and in vivo preclinical studies. Given the lack of alternative treatments in tuberculosis, the need to preserve our remaining antibiotics is paramount. These high stakes necessitate future efforts to support our preliminary work, making silibinin more clinically relevant to patients and healthcare professionals alike.

## Figures and Tables

**Figure 1 ijms-21-03714-f001:**
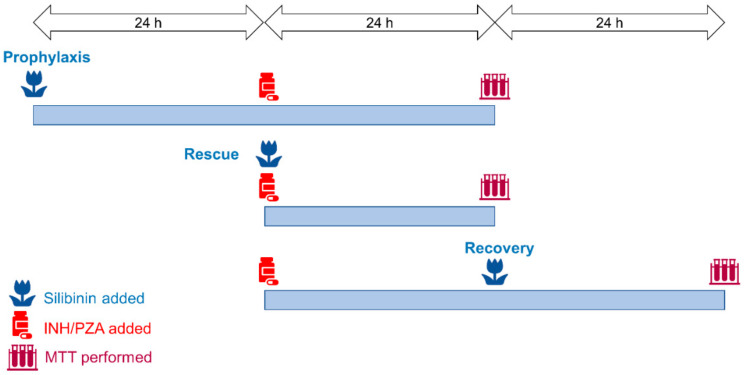
Treatment scheme involving silibinin’s role as a prophylactic, rescue, and recovery adjuvant. To simulate silibinin’s role as a preventive agent, silibinin was administered 24 h before the treatment with toxicants. To simulate silibinin’s role as rescue adjuvant, silibinin was co-administered with the toxicant regimen. To simulate silibinin’s role as a recovery adjuvant, silibinin was added 24 h after the toxicant regimen. The recovery experiments were further subdivided into two conditions: the first had a washout step, while the second did not have a washout step. In the simulation with washout, the toxicant regimen was replaced with silibinin alone and then treated for a further 24 h to investigate silibinin’s ability to aid patients in recovery after stopping the hepatotoxic regimen. In the simulation without washout, the toxicant regimen was replaced with a combination of silibinin and toxicant and treated for a further 24 h to investigate silibinin’s ability to mitigate further liver injury in patients who stay on the toxicant regimen. PZA, pyrazinamide; INH, isoniazid; MTT, 3-(4,5-dimethylthiazol-2-yl)-2,5-diphenyltetrazolium bromide.

**Figure 2 ijms-21-03714-f002:**
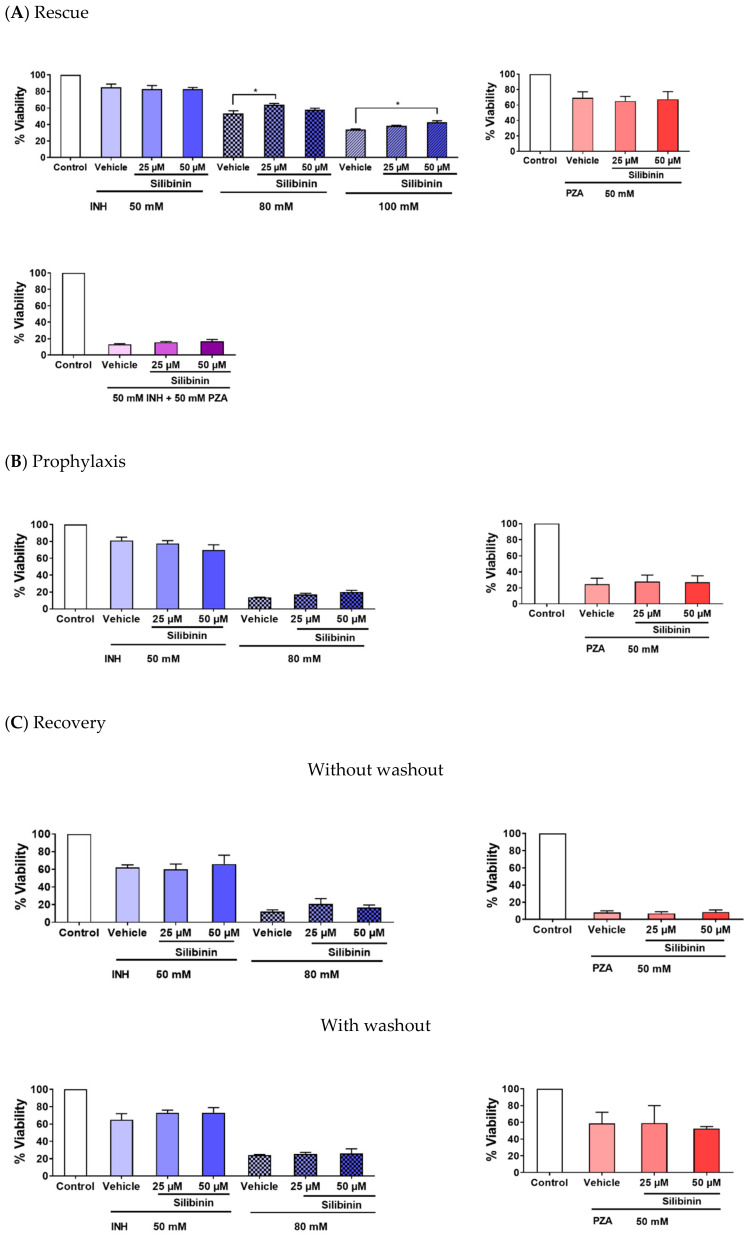
Silibinin mitigated isoniazid (INH)-induced hepatotoxicity, but not pyrazinamide (PZA)-induced hepatotoxicity. (**A**) Co-administration of silibinin at 25 µM reduced hepatotoxicity induced by 80 mM (one-way analysis of variance (ANOVA), *p* = 0.0231). Similarly, co-administration of silibinin at 50 µM reduced hepatotoxicity induced by 100 mM INH (one-way ANOVA, *p* = 0.0201). Co-administration of silibinin at either 25 or 50 µM, but did not reduce hepatotoxicity induced by 50 mM INH, 50 mM PZA, or a combination of INH and PZA (I/P) at 50 mM each (I/P 50/50). (**B**) Pre-administration of silibinin for 24 h, followed by the co-administration of silibinin with INH or PZA for a further 24 h, did not prevent hepatotoxicity induced by 50 mM INH, 80 mM INH, or 50 mM PZA. (**C**) Administration of INH or PZA for 24 h, followed by the administration of silibinin alone (with washout) or silibinin with INH or PZA (without washout) for a further 24 h, did not aid in the recovery of LO2 from 50 mM INH, 80 mM INH, or 50 mM PZA. Data represent mean ± S.E.M. of at least two replicates. * *p* < 0.05 vs. respective vehicle controls.

**Figure 3 ijms-21-03714-f003:**
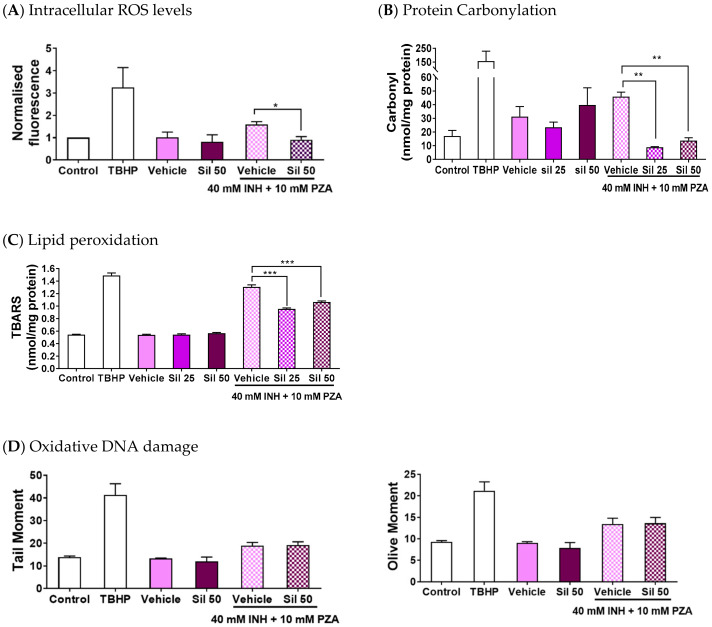
Silibinin reduced reactive oxygen species (ROS) levels and oxidative damage when co-administered with a combination of isoniazid (INH) and pyrazinamide (PZA). Positive controls were treated with the oxidising agent tert-butyl hydroperoxide (TBHP) 200 μM for 2 h. To avoid excessive hepatocyte death, the concentrations of INH and PZA were limited to 40 mM and 10 mM, respectively, when treated in combination (I/P 40/10) over 24 h. (**A**) 50 μM silibinin reduced intracellular ROS levels (t-test, *p* = 0.0466). (**B**) Silibinin decreased carbonylation levels, a marker of oxidative damage in proteins, at 25 μM (one-way ANOVA, *p* = 0.0015) and 50 μM (one-way ANOVA, *p* = 0.0023). (**C**) Silibinin reduced lipid peroxidation levels as measured by the thiobarbituric acid reactive substances (TBARS) assay at 25 μM (one-way ANOVA, *p* < 0.0001) and 50 μM (one-way ANOVA, *p* = 0.0007). (**D**) Silibinin’s reduction of ROS levels at 50 μM was independent of DNA oxidative damage reduction as visually assessed, and as measured quantitatively by tail moment and olive moments. Administration of silibinin alone did not trigger DNA fragmentation. Data represent mean ± S.E.M. of at least two replicates. * *p* < 0.05, ** *p* < 0.01, *** *p* < 0.001 vs. vehicle control co-administered with I/P 40/10.

**Figure 4 ijms-21-03714-f004:**
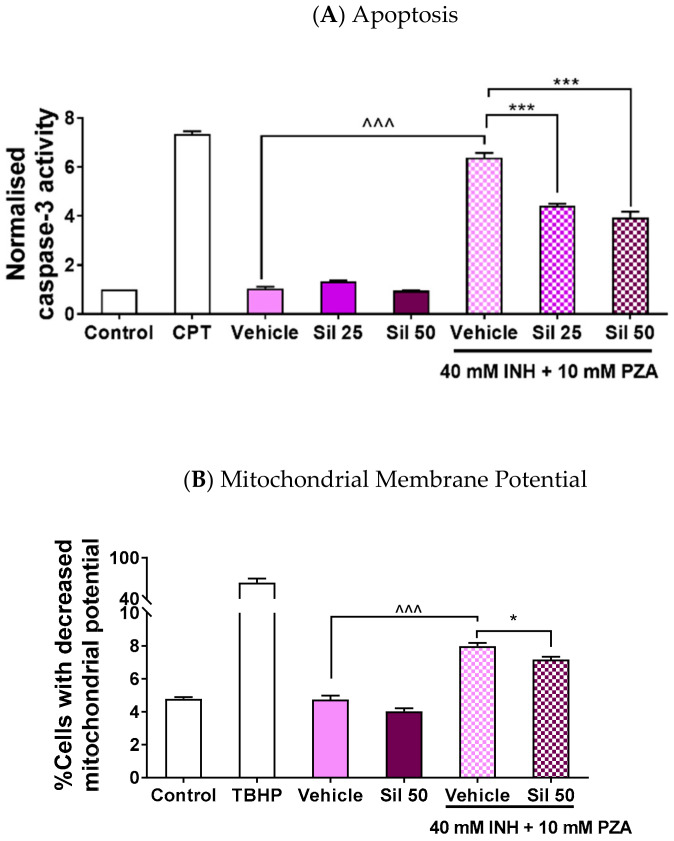
Silibinin reduced apoptosis when co-administered with a combination of INH and PZA by maintaining mitochondrial membrane potential. Various concentrations of silibinin were co-administered with I/P 40/10 over 18 h. (**A**) The administration of I/P 40/10 significantly increased caspase-3 activity (one-way ANOVA, *p* < 0.0001). The co-administration of silibinin with I/P 40/10 reduced the activity of caspase-3 when silibinin was administered at 25 μM (one-way ANOVA, *p* < 0.0001) and 50 μM (one-way ANOVA, *p* < 0.0001), suggesting that there was a reduction in apoptotic activity. The positive control was treated with camptothecin (CPT) 5 μM for 24 h. (**B**) The administration of I/P 40/10 negatively affected LO2 cells’ membrane potential (one-way ANOVA, *p* < 0.0001). Silibinin of 50 μM reduced the percentage of cells whose membrane potential was negatively affected by I/P 40/10 (one-way ANOVA, *p* = 0.0234). Positive control was treated with the oxidising agent TBHP 200 μM for 1 h. Data represent mean ± S.E.M. of three replicates. * *p* < 0.05, *** *p* < 0.001 vs. vehicle control co-administered with I/P 40/10, ^^^ *p* < 0.001 vs. respective vehicle controls.

**Figure 5 ijms-21-03714-f005:**
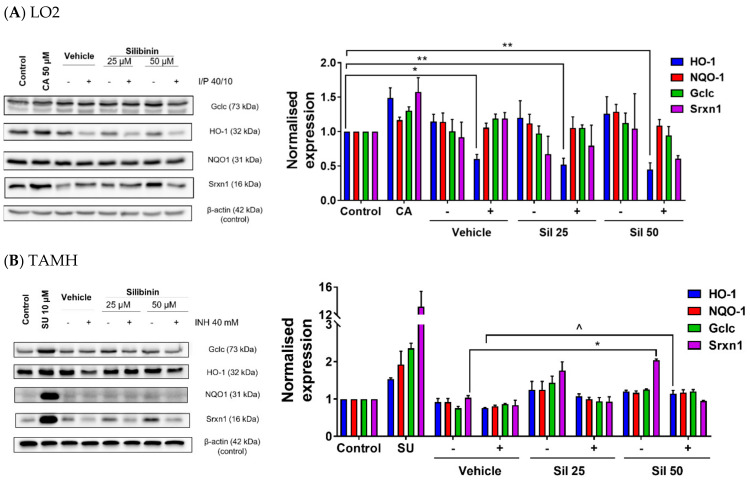
Silibinin induced expression of proteins in the nuclear factor (erythroid-derived 2)-like 2 (Nrf2)–antioxidant response element (ARE) pathway and restored protein expression. Vehicle control was treated with 0.05% *v*/*v* dimethyl sulfoxide (DMSO). ‘−’ denotes conditions without toxicant cotreatment; ‘+’ denotes conditions with toxicant cotreatment. (**A**) In LO2, silibinin’s reduction of ROS levels when co-administered with I/P 40/10 was independent of heme oxygenase-1 (HO-1) protein restoration. The administration of I/P 40/10 significantly reduced HO-1 levels without silibinin (one-way ANOVA, *p* = 0.0150), or with silibinin at 25 μM (one-way ANOVA, *p* = 0.0051) and 50 μM (one-way ANOVA, *p* = 0.0022). Positive controls were treated with the Nrf2–ARE inducer trans-cinnamaldehyde (CA) 50 μM for 24 h. Data represent mean ± S.E.M. of three replicates. * *p* < 0.05, ** *p* < 0.01 vs. negative control. (**B**) In transforming growth factor-α transgenic mouse hepatocytes (TAMH), 50 μM silibinin alone induced sulfiredoxin 1 (Srxn1) expression (one-way ANOVA, *p* = 0.0237), but the co-administration of silibinin with 40 mM INH did not restore Srxn1 expression to pre-suppression levels (one-way ANOVA, *p* = 0.0551). In contrast, 50 μM silibinin restored HO-1 expression (one-way ANOVA, *p* = 0.0333), but did not induce HO-1 when was administered alone (one-way ANOVA, *p* = 0.0564). This effect did not extend to glutamate-cysteine ligase catalytic subunit (Gclc) and NAD(P)H quinone dehydrogenase 1 (NQO1) restoration. Positive controls were treated with the Nrf2–ARE inducer sulphoraphane (SU) 10 μM for 24 h. Data represent mean ± S.E.M. of two replicates. * *p* < 0.05 vs. vehicle control, ^ *p* < 0.05 vs. vehicle control co-administered with hepatotoxic regimen.

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
