# Peer review of "An Evaluation of the In Vitro Roles and Mechanisms of Silibinin in Reducing Pyrazinamide- and Isoniazid-Induced Hepatocellular Damage"

_ijms, 2020, doi:10.3390/ijms21103714_

Round 1

Reviewer 1 Report

The manuscript by Goh et al. presents the role and mechanisms of silibinin in mitigating PZA- and INH-induced hepatotoxicity. The methods and results are adequate and well described, the conclusions drawn are supported by the data. I have just a few minor comments.

Lines 31-32: one in four, according to the latest estimation.

Line 37: it is usually called HRZE regimen

Please add ROS to the list of abbreviations

According to the literature, silibinin has already been used in clinical practice quite widely. Therefore, I’d suggest the authors elaborate on the clinical implications of their study. For example, there are reports on randomized trials assessing clinical efficacy and safety of Silibinin in tuberculosis patients (Gu et al., Int J Clin Exp Med. 2015).

Author Response

Response to Reviewer 1 Comments

The manuscript by Goh et al. presents the role and mechanisms of silibinin in mitigating PZA- and INH-induced hepatotoxicity. The methods and results are adequate and well described, the conclusions drawn are supported by the data. I have just a few minor comments.

Point 1: Lines 31-32: one in four, according to the latest estimation.

Response 1: We thank Reviewer 1 for highlighting this, and we have revised this statement in Lines 31-32.

Point 2: Line 37: it is usually called HRZE regimen.

Response 2: We thank Reviewer 1 for the suggestion. All references to the PIER regiment have been replaced with the HRZE regimen e.g. lines 37, 48, 73.

Point 3: Please add ROS to the list of abbreviations.

Response 3: We thank Reviewer 1 for highlighting this. ROS has now been added to the list of Abbreviations (line 553).

Point 4: According to the literature, silibinin has already been used in clinical practice quite widely. Therefore, I’d suggest the authors elaborate on the clinical implications of their study. For example, there are reports on randomized trials assessing clinical efficacy and safety of Silibinin in tuberculosis patients (Gu et al., Int J Clin Exp Med. 2015).

Response 4: We thank Reviewer 1 for the suggestion to elaborate on the clinical implications of our study. In addition to citing more in vitro, in vivo, and clinical studies in our manuscript, we have also emphasised the clinical implications in the following areas of our manuscript:

Lines 333-336 now discuss the possible clinical role of silibinin arising from its effect on stellate cells (stellate cells are gaining prominence in the literature for their role in liver disease): “While these two observations suggest that silibinin may not be involved in the protective or regenerative mechanisms that restore normal hepatocyte function after DILI has occurred [73], silibinin has been shown to reduce stellate cell migration [57,65,74], which plays an important role in mediating liver diseases involving fibrotic activity, liver injury, and liver regeneration [75].”

Lines 337-345 have now expanded to include clinical implications: “Therefore, future directions to better characterise silibinin’s role in recovery may centre around the use of co-cultures, which can mimic paracrine responses. These studies will help clinicians personalize regimens to patients’ conditions rapidly and accurately: instead of using a one-size-fit-all approach [76], regimens may be tailored to patients depending on their risk of DILI from genetic polymorphisms [68]. Ethical considerations in clinical practice often require investigators to exclude moderate-to-high-risk patients in their study [51], and consequently investigators may not recruit enough patients to power their study more effectively [49]. Our finding that silibinin was most effective in moderate-to-high DILI (Fig. 2A) may establish the moral basis for further clinical trials investigating silibinin’s hepatoprotective effect.”

Lines 360-362 now directly discuss silibinin’s safety profile, within the context of in vitro, in vivo, and clinical studies: “Coupled with the observation that silibinin did not induce caspase-3 activity (Fig. 4A), this reinforces silibinin’s clinical safety profile [51,78] and supports silibinin’s development in further studies.”

Lines 419-421 now directly reference the possibility that silibinin interacts through other injury pathways. Reviewer 2 has suggested that we list this as a potential limitation of our study: “One potential limitation of our study may lie with our finding that silibinin did not protect against PZA-induced hepatotoxicity may be attributed to other injury pathways [50]. Because these alternative pathways may only become apparent in vivo, they lie beyond the scope of our study.”

Reviewer 2 Report

The present study by Goh et al. reported the effectiveness of silibinin an herbal product on tuberculosis treatment (antitubercular regimen) and showed there result to be protective against drug-induced liver injury by using an in-vitro model. In this study, the author explored the role of silibinin’s in extenuating Pyrazinamide (PZA) and Isoniazid (INH) induced hepatotoxicity and reported a mechanistic approach for understanding silibinin’s hepatoprotective ability. The study result reported that silibinin preserved the viability of human fetal hepatocyte line LO2 when co-treated with 80 mM INH. Furthermore, the cell exhibited a lowering of apoptosis induced by a combination of 40 mM INH and 10 mM PZA that was suggested by decreased oxidative damage to mitochondria, proteins, and lipids. 

However, there are few comments to be responded before finalizing the decision on the MS. 

  1. Line 12, it is better to write in small letter drug-induced liver injury 
  2. Line 15, rewrites the sentence “The problem is compounded by the lack of safe and effective alternatives to the antitubercular regimen”.
  3. Line 54-56. The written lines are complicated to understand, as they were mixed up.
  4. Line 57, DNA need not to be abbreviated as it was used first time.
  5. In the last paragraph of introduction line 101-105, if the authors want, they can delete the lines, as you cannot put all information in the introduction section. I think you already describe comprehensibly introduction part.
  6. Line 123-125, what does this meaning.. did you take any sample from patients?
  7. Line 136, is the effect was synergistic?
  8. For figure 2, is the concentration was affordable to use it. You can use some paper as a reference that used similar viability reported or there is a need to show the significance level between the groups (Without washout section). 
  9. Line 183, how is the author could explain?
  10. Figure 4, section B, Explain the result for the Line 237-239.
  11. Line 254-256, What was the probable reason for this behavior or observation.
  12. Line 269, “Silibinin alone did not induce the expression of Gclc, HO-1, NQO1, and Srxn1” in what sense the author wrote the lines.
  13. Line 272-274, in the western blot it shows the expression of OH-1 in both condition
  14. Line 291-292, it is a limitation that the system needs to fully invivo study not only using in vivo marker
  15. Line 296, please mention here which table or figure is representing the aforementioned results
  16. Line 305-306 “Goldilocks zone” the explanation needs to be in the material and method section
  17. Line 311-313, So, at higher concentration it is toxic to the cells, it is better to mention here the exact dose.
  18. Line 314, is there was any study related to that, please mention here
  19. The result should be discussed somewhere else as the author discussed figure 5a. It is difficult to reach a coherent way of explanation. If you were providing any reason for the observation. better to rewrite the line 315-317
  20. Line 321, Did the author check any regenerative mechanism for this
  21. Line 326-328, No need to add up the sentence.
  22. Line 335, Do you have any reason for this behavior
  23. Line 377-380, many times the same sentences have been written. Therefore, it is better to delete it.  
  24. Line 403-404, the limitation of the study should be included.
  25. Line 421, As per the described experiment, the portion needs to be comprehensibly explored
  26. Line 473-474 the bracket was not closed.
  27. Several references were wrongly written, please modify them. Reference number 3, 28, 49, 50, 52,53, 57,58, 59,61 and 78

Author Response

Response to Reviewer 2 Comments

The present study by Goh et al. reported the effectiveness of silibinin an herbal product on tuberculosis treatment (antitubercular regimen) and showed there result to be protective against drug-induced liver injury by using an in-vitro model. In this study, the author explored the role of silibinin’s in extenuating Pyrazinamide (PZA) and Isoniazid (INH) induced hepatotoxicity and reported a mechanistic approach for understanding silibinin’s hepatoprotective ability. The study result reported that silibinin preserved the viability of human fetal hepatocyte line LO2 when co-treated with 80 mM INH. Furthermore, the cell exhibited a lowering of apoptosis induced by a combination of 40 mM INH and 10 mM PZA that was suggested by decreased oxidative damage to mitochondria, proteins, and lipids.

However, there are few comments to be responded before finalizing the decision on the MS.

Point 1: Line 12, it is better to write in small letter drug-induced liver injury.

Response 1: We thank Reviewer 2 for this suggestion. Drug-induced liver injury in line 12, as well as the entry in the Abbreviations section, have been edited.

Point 2: Line 15, rewrites the sentence “The problem is compounded by the lack of safe and effective alternatives to the antitubercular regimen”.

Response 2: We thank Reviewer 2 for this suggestion. We have changed line 15 to “Yet we lack safe and effective alternatives to the antitubercular regimen.”

Point 3: Line 54-56. The written lines are complicated to understand, as they were mixed up.

Response 3: We thank Reviewer 2 for the opportunity to clarify our manuscript. Our manuscript now reads as follows in lines 50-59:

“First, PZA and INH can be converted to reactive metabolites by drug metabolizing enzymes. PZA is oxidized to 5-hydroxypyrazinamide via xanthine oxidase; and both PZA and 5-hydroxypyrazinamide are further bioactivated by xanthine oxidase to the toxic metabolites, pyrazinoic acid and 5-hydroxypyrazinoic acid [17,18]. Between pyrazinoic acid and 5-hydroxypyrazinoic acid, 5-hydroxypyrazinoic acid may be the more toxic metabolite [18]. In contrast, INH can be activated to toxic metabolites via N-acetyltransferase [8,19,20], amidases [8,19,20], and CYP2E1 [8,19]. Notably, INH also induces CYP2E1 [21,22], thus increasing the rate at which INH is metabolised. These metabolites of INH increase the levels of intracellular ROS [8,23,24], which consequently damage vital cellular targets.”

Point 4: Line 57, DNA need not to be abbreviated as it was used first time.

Response 4: We thank Reviewer 2 for this suggestion, and we have included the abbreviation for Deoxyribonucleic acid (DNA) in line 59, as well as in the Abbreviations section.

Point 5: In the last paragraph of introduction line 101-105, if the authors want, they can delete the lines, as you cannot put all information in the introduction section. I think you already describe comprehensibly introduction part.

Response 5: We thank Reviewer 2 for this suggestion. We would like to keep the overview of our work so that readers can quickly and easily understand our study design and results.

Point 6: Line 123-125, what does this meaning.. did you take any sample from patients?

Response 6: We thank Reviewer 2 for the clarification. Our experiment looking at the potential effect of washout was conducted in vitro. There were two experiments. The first experiment involved a washout, in which the culture media containing the toxicants was replaced with fresh media. In the second experiment without a washout, the culture media continued to contain the toxicants.

To clarify this meaning for our readers, the relevant section (lines 123-130) now reads as follows:

“In our exploration of silibinin as a recovery adjuvant, we investigated silibinin’s ability to mitigate hepatocyte toxicity in vitro after hepatotoxic induction. We set up a pair of experiments to demonstrate this in vitro. In the first experiment, the hepatocytes underwent a washout procedure, where we replaced the toxicant media with fresh media after 24 h. In the second experiment, there was no washout and the toxicant remained in the culture medium. By comparing silibinin’s in vitro hepatoprotective ability between this pair of experiments, we simulated silibinin’s potential ability to mitigate further liver injury in patients who either discontinue or stay on the hepatotoxic regimen.”

Related to this clarification on our experimental design, we have also incorporated point 25 highlighted by Reviewer 2 into the Methods section by elaborating on it in lines 441-450.

Point 7: Line 136, is the effect was synergistic?

Response 7: We thank Reviewer 2 for the suggestion of a synergistic effect between INH and PZA. This possibility has now been included in lines 140-144:

“Interestingly, silibinin’s protection against INH-induced hepatotoxicity also seemed to decrease when silibinin was administered together with a combination of 50 mM INH and 50 mM PZA (Fig. 2A). This observed combinatorial toxic effect between INH and PZA is reminiscent of the synergistic toxicity between the four drugs found in the HRZE regimen that has been reported in vitro [29,55,56], in vivo [57,58], and in humans [59,60].”

Point 8: For figure 2, is the concentration was affordable to use it. You can use some paper as a reference that used similar viability reported or there is a need to show the significance level between the groups (Without washout section).

Response 8: We thank Reviewer 2 for this suggestion. To strengthen the rationale for our use of concentrations, we have edited lines 115-117 to read:

“Similarly, to optimise the concentration windows of INH and PZA, we determined their IC50s to be 73 and 60 mM respectively (Fig. S2), observations consistent with reports in those made in HepaRG [47] and HepG2 [29].”

Point 9: Line 183, how is the author could explain?

Response 9: We thank Reviewer 2 for the suggestion to elaborate on this process of quantitation. The following line has been added for further clarity in lines 189-190: “Specifically, we quantified the oxidative damage through protein carbonylation levels, lipid peroxidation levels, and DNA fragmentation.”

Point 10: Figure 4, section B, Explain the result for the Line 237-239.

Response 10: We thank Reviewer 2 for the suggestion to elaborate on this phenomenon. We have elaborated on this in lines 204-207:

“As caspase-3 is the final mediator of both the intrinsic and extrinsic apoptotic pathways, silibinin’s mitigation of caspase-3 induction suggests that it reduces the overall apoptotic activity induced by I/P 40/10 administration.”

Point 11: Line 254-256, What was the probable reason for this behavior or observation.

Response 11: We thank Reviewer 2 for probing the reason underlying the suppression of HO-1 by INH, and not PZA. In lines 62-70, we wrote that the exact mechanisms underlying the suppression of the Nrf2 antioxidant pathway have not been fully elucidated, despite two studies showing how PZA and INH may increase cellular susceptibility to oxidative stress by downregulating downstream antioxidant proteins i.e. Gclc, NQO1, HO-1, and Srxn1. This formed the rationale for our investigation of the effects of INH and PZA on Gclc, NQO1, HO-1, and Srxn1.

“Second, PZA and INH suppress the Nuclear factor (erythroid-derived 2)-like 2 (Nrf2)-antioxidant response element (ARE) pathway that protects cells from oxidative damage [27,28]. Consequently, both INH [27] and PZA [28] increase cellular susceptibility to oxidative stress by decreasing the expression of downstream antioxidant proteins; the antioxidant proteins affected include Glutamate-cysteine ligase catalytic subunit (Gclc), NAD(P)H quinone dehydrogenase 1 (NQO1), Heme oxygenase-1 (HO-1), and Sulfiredoxin 1 (Srxn1). Indeed, RMP, INH, and PZA have been shown to potentiate the hepatotoxic effects of one another in in vitro assays involving HepG2 [29], though the underlying mechanisms of hepatotoxicity caused by these antitubercular drugs remain poorly understood to date. Seen in totality, the broad strokes illustrated by these studies denotes the need for hepatoprotective strategies to counter the increase in oxidative stress induced by these drugs.”

In lines 369-371, we cited two other studies in discussing our finding that INH inhibits the HO-1 pathway. Specifically, these studies found that “INH reduces both the mRNA transcription levels and the activity of HO-1 [27,28]”, thus suggesting that there is an overall downregulation of HO-1 transcription and translation.

In contrast to INH, the effect of PZA on the Nrf2-ARE pathway has been less explored in the literature. To date, PZA has been reported to inhibit basal and inducible ARE activity in 3T3-L1 and HaCaT cells [28]. Considering how 3T3-L1 and HaCaT cells are mouse-derived adipose cells and human skin cells respectively, our observations in LO2 can be explained in the context of the “innate metabolic, transporter-related, and physiological differences between various cell lines” (lines 394-395).

Another reason for why INH, rather than PZA, suppresses HO-1 can also be gleaned from 396-401, where we offer a broader justification underscoring the importance of using LO2 cells compared to other liver and non-liver cell lines. By elaborating on how the use of LO2 cells better contextualises the use of silibinin within HRZE-induced injury, we hypothesise that the different metabolic activity observed in LO2 cells may also play a role in the observed differences between our findings and those reported in the literature.

Point 12: Line 269, “Silibinin alone did not induce the expression of Gclc, HO-1, NQO1, and Srxn1” in what sense the author wrote the lines.

Response 12: We thank Reviewer 2 for the clarification. We observed that the treatment of cells with silibinin alone, i.e. without further addition of toxicant, did not result in the induction in any of these four markers of Gclc as represented by the bars grouped in the sections labelled ‘-’ (Fig. 5A). We wanted to highlight that silibinin alone does not perturb oxidative stress response and is only of significance when LO2 suffers an insult, and that this might also be linked to silibinin’s safety.

To improve on the clarity of this sentence, we have discussed this point in our main text (lines 266-268):

“Silibinin alone did not induce the expression of Gclc, HO-1, NQO1, and Srxn1, suggesting that it does not perturb endogenous oxidative stress response, a finding indicative of silibinin’s safety.”

Point 13: Line 272-274, in the western blot it shows the expression of OH-1 in both condition

Response 13: We thank Reviewer 2 for this comment. Fig. 5A indeed demonstrates that HO-1 is indeed expressed under all conditions. The reduction in HO-1 expression is especially evident in the case of treatment with INH (lanes marked with ‘+’), which suggests that INH significantly reduces HO-1 expression.

In lines 369-371, we further elaborate on this finding and show how it corroborates the extant literature and mechanism by which INH suppresses HO-1 expression:

“The observation that INH suppressed HO-1 expression in LO2 when used in combination with PZA (Fig. 5A) is consistent with the current paradigm in which INH reduces both the mRNA transcription levels and the activity of HO-1 [27,28].”

Point 14: Line 291-292, it is a limitation that the system needs to fully invivo study not only using in vivo marker

Response 14: We thank Reviewer 2 for this thoughtful suggestion. Indeed, in our conclusion (lines 532-534), we show how the findings in our manuscript help to complement findings in past in vitro studies, and reconcile them with future in vivo preclinical studies. Therefore, the value of our manuscript derives from the following two aspects.

First, our model establishes a proof of concept that forms the rational and moral basis for future in vivo studies of silibinin in animals. We have demonstrated that silibinin may have a hepatoprotective effect in vitro, which arises from two mechanisms: silibinin’s ability to 1) reduce levels of reactive oxygen species that cause oxidative stress, and 2) induce endogenous antioxidant levels that protect against oxidative stress.

Second, our study reconciles the differences observed in previous in vitro studies. In lines 362-381 (now rephrased, with thanks to Reviewer 2’s suggestions), we compare LO2 to TAMH to reconcile the mechanisms underlying the differences in the endogenous antioxidant pathway between them. In the next paragraph (lines 382-389), we compared our findings with other in vitro studies discussed in the literature.

Point 15: Line 296, please mention here which table or figure is representing the aforementioned results

Response 15: We thank Reviewer 2 for this suggestion. Lines 303-305 now read as follows:

“First, silibinin reduces intracellular levels of oxidative stress and oxidative damage to intracellular targets (Fig. 3A-D) and mitochondria (Fig. 4B), leading to decreased apoptotic activity (Fig. 4A).”

Point 16: Line 305-306 “Goldilocks zone” the explanation needs to be in the material and method section

Response 16: We thank Reviewer 2 for the request to clarify how the “Goldilocks zone” was derived. As part of our response to Point 8 raised by Reviewer 2, we have defined the “Goldilocks zone” to be a range of concentrations which are neither too high nor too low, over which we hypothesised that silibinin would exhibit its hepatoprotective effect.

The “Goldilocks zone” is a hypothesis derived from an empirical observation (Fig. S3) and was not theoretically derived; we have described the experimental setup in section 4.2 Cell viability assay (lines 433-450).

Point 17: Line 311-313, So, at higher concentration it is toxic to the cells, it is better to mention here the exact dose.

Response 17: We thank Reviewer 2 for the suggestion to reinforce this section using our observation that silibinin is toxic at concentrations beyond 100 µM (Fig. S1B).

As the optimal silibinin concentration is discussed in lines 323-327, we have decided to incorporate Reviewer 2’s suggestion there instead:

“The optimal silibinin concentration from our viability experiments was determined to be 25 μM when DILI was induced with 80 mM INH (Fig. 2A). This may be explained by silibinin’s pro-oxidative and pro-apoptotic effects at higher concentrations (i.e. beyond 100 µM) (Fig. S1B), an observation that corroborates experiments conducted by other research groups in rats [71] and other in vitro cell lines [31,47,72].”

Point 18: Line 314, is there was any study related to that, please mention here

Response 18: We thank Reviewer 2 for this suggestion to include other studies. This sentence, now in line 328, is derived from our argument laid out in previous sentences. Therefore, our statement that “increasing silibinin’s concentration may not always lead to increased hepatoprotection”, reconciles our observations in Fig. 2A and Fig. S1B with the studies listed in lines 326-327 i.e. [65] and [30,46,66]. Therefore, we have rewritten it for greater clarity (lines 326-327):

“Our results and literature reports [31,47,71,72] thus suggest that increasing silibinin’s concentration may not always lead to increased hepatoprotection.”

Point 19: The result should be discussed somewhere else as the author discussed figure 5a. It is difficult to reach a coherent way of explanation. If you were providing any reason for the observation. better to rewrite the line 315-317

Response 19: We thank Reviewer 2 for this suggestion. As we have elaborated on this issue in previous paragraphs, we have summarised our results accordingly.

Point 20: Line 321, Did the author check any regenerative mechanism for this

Response 20: We thank Reviewer 2 for this question. Given that our cell viability assays suggest that silibinin did not seem to promote recovery in our in vitro model, we considered identifying the potential regenerative mechanism underlying silibinin’s hepatoprotective effect.

As most regenerative studies centre around the use of animal models that can recapitulate the liver microenvironment, we hypothesise that regenerative studies may be best detected in these situations. Our hypothesis is further supported by silibinin’s ability to “reduce stellate cell migration [57,65,74], which plays an important role in mediating liver diseases involving fibrotic activity, liver injury, and liver regeneration [75].”

We have also incorporated comments from Reviewer 1 in rewriting these lines in the manuscript.

Point 21: Line 326-328, No need to add up the sentence.

Response 21: We thank Reviewer 2 for this suggestion to make our writing more concise. In line with advice from both Reviewer 1 and Reviewer 2, we have rewritten this paragraph.

Point 22: Line 335, Do you have any reason for this behaviour

Response 22: We thank Reviewer 2 for the suggestion to discuss our hypothesis of the cause underlying this observation. We attribute this observation to how silibinin “may not protect against all forms of INH- and PZA-induced oxidative damage” (lines 355-356).

Furthermore, our study focused on study of silibinin in human-relevant LO2 human foetal hepatocytes in vitro, while silibinin protects against doxorubicin-induced DNA fragmentation in mouse liver cells in vivo, further alluding to our hypothesis that “silibinin’s induction of the Nrf2-ARE pathway may contribute towards its hepatoprotective effect in rodents” (lines 377-378) may differ from the mechanism underlying its hepatoprotective effect in humans.

Point 23: Line 377-380, many times the same sentences have been written. Therefore, it is better to delete it. 

Response 23: We thank Reviewer 2 for this suggestion to make our writing more concise. The relevant paragraph has been removed. The relevant effects are now discussed in the paragraphs describing silibinin’s effect on LO2 (lines 363-374) and TAMH (lines 375-382).

Point 24: Line 403-404, the limitation of the study should be included.

Response 24: We thank Reviewer 2 for this suggestion. Lines 419-421 now read: “One potential limitation of our study may lie with our finding that silibinin did not protect against PZA-induced hepatotoxicity may be attributed to other injury pathways [50]. Because these alternative pathways may only become apparent in vivo, they lie beyond the scope of our study.”

Point 25: Line 421, As per the described experiment, the portion needs to be comprehensibly explored

Response 25: We thank Reviewer 2 for the suggestion to flesh out our Methods. We have now done so in lines 442-451:

“In measuring the toxicity of silibinin, INH, and PZA on cells, a series of concentrations was prepared in media and administered to silibinin for 24 h, after which the cell viability was measured. To simulate silibinin’s role as a preventive agent, the cells were treated with silibinin for 24 h, then with both silibinin and the toxicants for a further 24 h before cell viability was assessed. To simulate silibinin’s role as a rescue adjuvant, the cells were treated with both silibinin and the toxicant for 24 h before cell viability was assessed. To simulate silibinin’s role as a recovery adjuvant without washout, the cells were treated with the toxicant for 24 h, then with silibinin and the toxicants for a further 24 h before cell viability was assessed. To simulate silibinin’s role as a recovery adjuvant with washout, the cells were treated with the toxicant for 24 h, then with only silibinin for a further 24 h before the cell viability was assessed.”

Point 26: Line 473-474 the bracket was not closed.

Response 26: We thank Reviewer 2 for highlighting this to us. We have now closed the bracket in line 503. The sentence now reads:

“300 µL of protease assay buffer (2 mM DTT, 10% v/v glycerol, 20 mM HEPES, 20 mM Ac-DEVD-AMC Caspase-3 Fluorogenic Substrate (BD Pharmingen, United States)) was added and the samples were incubated at 37 °C in the dark for 1 h.”

Point 27: Several references were wrongly written, please modify them. Reference number 3, 28, 49, 50, 52,53, 57,58, 59,61 and 78

Response 27: We thank Reviewer 2 for highlighting this to us. We have ensured that the correct citations and references in the bibliography have been included and are formatted in line with IJMS’ style described in IJMS’ Guide for Authors.

Round 2

Reviewer 2 Report

The author addressed all concerns.